# Immunogenicity and Safety of Omicron-Containing Multivalent COVID-19 Vaccines in Unvaccinated and Previously Vaccinated Adults

**DOI:** 10.3390/vaccines12101109

**Published:** 2024-09-27

**Authors:** Suad Hannawi, Alaa Abuquta, Linda Saf Eldin, Aala Hassan, Ahmad Alamadi, Cuige Gao, Adam Abdul Hakeem Baidoo, Xinjie Yang, Huo Su, Jinxiu Zhang, Liangzhi Xie

**Affiliations:** 1Internal Medicine Department, Al Kuwait-Dubai (ALBaraha) Hospital, Dubai 00000, United Arab Emirates; suad1@ausdoctors.net (S.H.); alaasalah19870@gmail.com (A.H.); 2Accident and Emergency Department, Al Kuwait-Dubai (ALBaraha) Hospital, Dubai 00000, United Arab Emirates; alaa.quta86@gmail.com; 3General Surgery Department, Al Kuwait-Dubai (ALBaraha) Hospital, Dubai 00000, United Arab Emirates; drlindasafeldin@gmail.com; 4Ear, Nose and Throat Department (ENT), Al Kuwait-Dubai (ALBaraha) Hospital, Dubai 00000, United Arab Emirates; ahmad.alamadi@ehs.gov.ae; 5Beijing Engineering Research Center of Protein and Antibody, Sinocelltech Ltd., Beijing 100176, China; cuige_gao@sinocelltech.com (C.G.); baidooadam@sinocelltech.com (A.A.H.B.); xinjie_yang@sinocelltech.com (X.Y.); huo_su@sinocelltech.com (H.S.); jinxiu_zhang@sinocelltech.com (J.Z.); 6Cell Culture Engineering Center, Chinese Academy of Medical Sciences & Peking Union Medical College, Beijing 100005, China

**Keywords:** immunogenicity, safety, multivalent vaccine, SARS-CoV-2, Omicron

## Abstract

The SARS-CoV-2 evolution trajectory remains uncertain, and the antigenic characteristics of future variants are highly unpredictable. We report the immunogenicity and safety of multivalent COVID-19 vaccines, SCTV01E and SCTV01E-1, against Omicron BA.5. This phase 2 trial randomized 400 adults into two cohorts, 160 unvaccinated (3 doses) and 240 previously vaccinated (2 doses) individuals to receive 30 µg SCTV01E-1 or 30 µg SCTV01E (1:1) between 4 November and 28 November 2022. Among the unvaccinated cohort, day 42 geometric mean fold rises (GMFRs) of neutralizing antibodies (nAb) against Omicron BA.5 were reported to be 12.8× and 20.5× over day 0 for SCTV01E-1 and SCTV01E, respectively. On day 178, both vaccines increased geometric mean titers (GMTs) of nAb against BA.5 following the booster dose compared to pre-booster levels on D150. Similar frequencies of solicited [6.2% (5/81) and 7.6% (6/79)] and unsolicited [11.1% (9/81) and 10.1% (8/79)] adverse events (AEs) were reported in SCTV01E-1 and SCTV01E groups, respectively. Grade 3 or more AEs were < 2% in both vaccine groups [SCTV01E-1: 1.2% (1/81), SCTV01E: 1.3% (1/79)]. In the previously vaccinated cohort, similar GMFRs were reported on day 28 (SCTV01E-1: 9.4× and SCTV01E: 8.7×) over baseline (D0). On day 148, both vaccines showed increased nAb levels with similar GMFRs over D120. Comparable incidences of solicited [13.2% (16/121) and 10.9% (13/119)] and unsolicited [17.4% (21/121) and 10.9% (13/119)] AEs were reported in SCTV01E-1 and SCTV01E groups, respectively. Numerically identical ≥ grade 3 AEs [SCTV01E-1: 1.7% (2/121) and SCTV01E: 1.7% (2/119)] were reported. This trial demonstrates the effectiveness of updated multivalent vaccines with acceptable safety profiles.

## 1. Introduction

The dynamic evolution of SARS-CoV-2 infection has resulted in several new variants, with Omicron and its overlapping sublineages being the latest (in a long list of variants “clan”) [1,2,3]. Large-scale vaccination programs have been widely adopted as the go-to approach to mitigate the coronavirus disease-2019 (COVID-19) burden through the induction of wide-scale population immunity with considerable success [4]. However, the overlapping Omicron subvariants in circulation have created a complex, unrelenting COVID-19 epidemiological landscape of multiple antigenically related yet distinct viral mutations armed with immune escape abilities, an alarming trend that raises concerns about the protective effect of first-generation COVID-19 vaccines against Omicron infections [5,6,7].

To stay alert to the looming threat, the World Health Organization (WHO), on 5 May 2023, declared that COVID-19 is now an established and ongoing health issue [8]. A month later, on 15 June 2023, the U.S. Food and Drug Administration (FDA) organized the 182nd Vaccines and Related Biological Products Advisory Committee (VRBPAC) meeting, where the periodic update of COVID-19 vaccine strain composition for future vaccination campaigns received full endorsement [9]. The shift in vaccination strategy towards updated COVID-19 vaccines is necessary to match the pace of the SARS-CoV-2 viral evolution to protect the general population from (mixed) SARS-CoV-2 variant re-infection.

At the peak of the Omicron BA.4/5 wave [10], we designed this study to investigate the safety and immunogenicity of SCTV01E-1 with SCTV01E as a control in two cohorts: COVID-19 vaccine-naive adults and those who had received two or three doses of inactivated COVID-19 vaccine. SCTV01E is a tetravalent SARS-CoV-2 protein vaccine with spike proteins from four variants (Alpha, Beta, Delta, and Omicron BA.1). The antigens were produced using stable CHO cell lines, purified, and combined with a squalene-based adjuvant (SCT-VA02B) in a single vial [11]. SCTV01E-1 protein vaccine composed of Alpha, Beta, Delta, and Omicron (BA.1/BA.4/BA.5) antigens followed a similar design and manufacturing process. SCTV01E [Alpha/Beta/Delta/Omicron (BA.1)] and SCTV01E-1 [Alpha/Beta/Delta/Omicron (BA.1/BA.4/5)] have similar antigen composition except for the BA.4/5 antigen strain. SCTV01E-1 and SCTV01E vaccines are intended for routine vaccination in at-risk populations and addressing new variants.

We have previously reported 28 days post-vaccination clinical immunogenicity data of SCTV01E, which demonstrated that a single dose of SCTV01E could induce significantly high neutralizing antibodies (nAb) and seroresponse rates (SRR) against Omicron BA.1 and BA.5 in adults previously vaccinated with monovalent mRNA vaccine [12] and inactivated COVID-19 vaccine immunization [13]. SCTV01E’s prototype vaccine, SCTV01C (bivalent protein vaccine), was administered as a two-dose primary series in COVID-19 vaccine-naïve adults in China, and markedly high levels of spike-IgG and nAb responses to SARS-CoV-2 variants were observed [14]. SCTV01E and SCTV01C received authorization for Emergency Use in China from the National Medicinal Product Agency (NMPA) of China on 4 December 2022 and 22 March 2023, respectively.

This report presents clinical immunogenicity and safety data following a 3-dose regimen in COVID-19 vaccine naïve and a 2-dose scheme in previously vaccinated adults.

## 2. Materials and Methods

### 2.1. Trial Design

This randomized, double-blinded, positive-controlled phase 2 study evaluated the immunogenicity and safety of an Omicron-containing SCTV01E-1 COVID-19 vaccine, with SCTV01E as a comparator. This trial was conducted at the Al Kuwait Hospital in Dubai, United Arab Emirates (UAE), and comprised Cohort 1—unvaccinated adults and Cohort 2—those with a previous COVID-19 vaccination history. The trial was registered at ClinicalTrials.gov number, NCT 05522829; and followed the ethical requirements of Good Clinical Practice and the Declaration of Helsinki. The UAE Ministry of Health and Prevention (Reference number: RCMOHP/CT2/0139/2022) approved the protocol, informed consent, and study amendments.

### 2.2. Study Participants

Eligibility criteria included male or female adults aged ≥18 years with the key target population: Cohort 1—COVID-19 vaccine naïve population; Cohort 2—population previously fully vaccinated (2 or 3 doses) with inactivated COVID-19 vaccine, with a 3 to 24-month interval between the last dose of their previous vaccination and the first study vaccination. All Cohort 1 adults did an anti-SARS-CoV-2 nucleocapsid antibody test at screening. The exclusion criteria for both cohorts included individuals with a positive RT-PCR for COVID-19 at screening, recent fever, history of allergic reactions to vaccines or drugs, and history of certain infections or diseases. Additionally, Cohort 2 adults with a known history of COVID-19 within 6 months before screening and a history of drugs or vaccines used to prevent COVID-19 were excluded. Enrollment was voluntary, and all participants provided written informed consent before any study procedure. Refer to the study protocol for full details on inclusion and exclusion criteria.

### 2.3. Randomization and Masking

An independent statistician from a third party generated randomization codes following a successful eligibility assessment. In Cohort 1, participants were randomized 1:1 to receive 3 doses of either 30 µg SCTV01E-1 or 30 µg SCTV01E. Likewise, participants in Cohort 2 were randomized 1:1 to receive 2 doses of either 30 µg SCTV01E-1 or 30 µg SCTV01E. Allocation was stratified based on age, anti-SARS-CoV-2 nucleocapsid antibody test results, the interval between the last and study vaccinations, and the number of previous doses. Blinding was maintained for all involved parties.

### 2.4. Procedures

Both SCTV01E-1 and SCTV01E were supplied in single-use vials as a sterile, emulsified, white solution, 0.5 mL/vial, stored and transported at 2–8 °C protected from light, with a validity period of 24 months. During the screening visit (baseline, day 0), all participants (Cohorts 1 and 2) underwent a complete physical examination before vaccination and provided blood samples for baseline immunogenicity testing. Notably, only Cohort 1 participants did an anti-SARS-CoV-2 nucleocapsid antibody test at screening (before first vaccination). Well-trained trial staff administered study vaccines intramuscularly into the participant’s upper arm.

### 2.5. Immunogenicity and Safety Assessments

Immunogenicity assessment samples were collected on pre-specified days to evaluate the nAb levels against Omicron BA.5: Cohort 1 samples were on days 0 (before the first dose), 42 (14 days post-second dose), 150 (before the third dose), and 178 (28 days post-third dose). Cohort 2 samples were collected on days 0 (before the first dose), 28 (28 days post-first dose), 120 (before the second dose), and 148 (28 days post-second dose). Biogenix (Biogenix, Abu Dhabi, United Arab Emirates) and G42 LABORATORY LLC (Dubai, United Arab Emirates) verified and performed the plaque reduction neutralization test (PRNT50) assay per the European Medicines Agency (EMA) and the US-FDA guidelines on biomarker assays. Safety assessment post-study vaccination for both Cohorts 1 and 2 included solicited AEs within 7 days, unsolicited AEs within 28 days, serious adverse events (SAEs), and adverse events of special interest (AESIs) throughout the entire study period. We used FDA toxicity criteria in preventive vaccine clinical trials to grade AEs.

### 2.6. Study Outcomes

The primary endpoints were geometric mean titers (GMT) of nAb against Omicron BA.5: Cohort 1—days 42 (14 days post-second vaccination) and 178 (28 days post-third dose); Cohort 2—days 28 and 148 (28 days post-second dose). The secondary immunogenicity endpoints included seroresponse rates of nAb against Omicron BA.5: Cohort 1 on days 42 and 178; Cohort 2 on days 28 and 148. Seroresponse is defined as achieving a level equal to or above the lower limit of quantitation (LLOQ) for participants with baseline nAb level < LLOQ or showing more than a 4-fold rise if the baseline levels are already at or above LLOQ. The safety secondary endpoints for both cohorts were the incidences and severities of solicited AEs within 7 days, unsolicited AEs within 28 days, and SAEs and AESIs throughout the entire study.

### 2.7. Statistical Analysis

The design was to achieve at least 80% power to demonstrate that SCTV01E-1 was superior to SCTV01E in terms of neutralizing antibodies against Omicron BA.5 with a one-sided type 1 error of 0.025. Cohort 1 sample size was determined based on the following assumptions: the ratio nAb levels (GMT) of SCTV01E-1 versus SCTV01E ≥ 1.7; the standard deviation of nAb under log10 transformation is 0.45; the dropout rate is about 10%. Similarly, Cohort 2 assumptions: nAb GMT ratio of SCTV01E-1 vs. SCTV01E ≥ 1.5; the standard deviation of nAb under log10 transformation is 0.45; the dropout rate is about 10%. For both cohorts, the safety analysis population included all individuals who received at least one study vaccine. The primary outcome included solicited (local and systemic) and unsolicited AEs within 7 days and 28 days, respectively, after the study vaccination. We have reported the proportion of participants with at least one solicited AE of Grade ≥ 3 for each group. We used the Medical Dictionary for Regulatory Activities (MedDRA), version 25.1, to code unsolicited AEs, categorizing them by system organ class (SOC) and preferred term (PT). The number and percentage of participants who experienced at least one event for each AE (solicited or unsolicited) are presented in this report.

Immunogenicity analysis was based on the immunogenicity per-protocol set (i-PPS), which randomized participants who received the planned vaccination per schedule and had no significant protocol deviations, with a valid immunogenicity test result at baseline and at least one accurate result after receiving the study vaccine. The primary immunogenicity outcome of nAb against the Omicron BA.5 was analyzed and reported as GMT with a corresponding 2-sided 95% CI estimated at each post-baseline time point based on a log-transformed scale. Comparison of the least squares geometric mean titers (LS GMT) of nAb between the treatment groups was performed at each post-baseline time point. Seroconversion and seroresponse between the treatment groups were compared using the Cochran–Mantel–Haenszel method stratified by stratification factors used in the randomization.

## 3. Results

### 3.1. Study Participants, Demographics and Baseline Characteristics

Four hundred healthy adults who met the eligibility criteria between 4 November 2022 and 28 November 2023 were enrolled into two cohorts to receive either 30 µg SCTV01E-1 or 30 µg SCTV01E in a 1:1 ratio (Figure 1). Cohort 1 comprised 160 unvaccinated adults who received at least one dose of study vaccines (SCTV01E-1: 81 and SCTV01E: 79), of which 1.9% (3/160) had prior medical or surgical history. Cohort 2 comprised 240 adults with inactivated COVID-19 vaccination history (SCTV01E-1: 121 and SCTV01E: 119), of which 2.5% (6/240) reported medical or surgical history. Demographics and baseline clinical characteristics were comparable across vaccine groups in both cohorts (Table 1 and Table 2). In Cohort 1, no participants reported prior SARS-CoV-2 infection; however, 38.1% (61/160) tested positive for anti-nucleocapsid antibody at baseline before the first study vaccination. In Cohort 2, only 3.3% (8/240) reported a known history of previous SARS-CoV-2 infection. All 240 participants had received either two doses (90%) or three doses (10%) of the inactivated COVID-19 vaccine. Trial participants’ previous vaccination history distribution was 70.4% (169/240) and 29.6% (71/240) for Sinopharm inactivated COVID-19 vaccine and CoronaVac (Sinovac inactivated COVID-19 vaccine), respectively. The interval between investigational and prior COVID-19 vaccination was 3–5 months (0.4%) and 6–24 months (99.6%), respectively. Full details are in Table 1 and Table 2.

### 3.2. Immunogenicity

Immune responses were examined on days 42 (14 days after the second dose) and 178 (28 days after the third dose) for Cohort 1 and on days 28 (after the first dose) and 148 (28 days after the second dose) for Cohort 2.

#### 3.2.1. Cohort 1

##### Day 42 GMTs of Live Virus nAb and Seroresponse RATES against Omicron BA.5

For the primary endpoint on day 42 (14 days after the second vaccination), the available immunogenicity data per i-PPS (immunogenicity per protocol set) were for 117/160 participants: 55 in the SCTV01E-1 group and 62 in the SCTV01E group. Post-second vaccination, the GMTs of nAb against Omicron BA.5 significantly increased over baseline (Figure 2) in both SCTV01E-1 and SCTV01E groups. The GMTs (95% CI) of nAb against Omicron BA.5 were 4930 and 4049 with corresponding GMFRs of 12.8× (95% CI: 9.0, 18.0) and 20.5× (95% CI: 13.9, 30.1) from baseline (D0) for SCTV01E-1 and SCTV01E, respectively. The least squares (LSs) geometric mean ratio (GMR) between the SCTV01E-1 group and the SCTV01E group was 1.17 (95% CI: 0.86, 1.61), with the LS GMR 95% LCI > 0.67, which demonstrated that SCTV01E-1 was non-inferior to SCTV01E. Seroresponse of live virus nAb against Omicron BA.5 on day 42 was seen in 89.1% (49/55) of participants for the SCTV01E-1 group and 88.7% (55/62) of participants for the SCTV01E group, with no significant difference observed between the two groups (*p*-value = 0.8594) (Appendix A).

##### Day 178 GMTs of Live Virus nAb and Seroresponse Rates against Omicron BA.5

On day 178 (28 days after the third dose), available immunogenicity data per i-PPS were from 75 participants, with 36 and 39 in SCTV01E-1 and SCTV01E groups, respectively (Figure 2). GMTs of nAb targeting Omicron BA.5 were 3551 and 2606 reported for SCTV01E-1 and SCTV01E, respectively, with numerically equivalent GMFRs (95% CI) of 1.4 times compared to day 150 (before the third dose) observed for both SCTV01E-1 and SCTV01E groups. The LS GMR (95% CI) between SCTV01E-1 and SCTV01E groups was 1.31 (0.90, 1.91), with the LS GMR 95% LCI > 0.67, indicating non-inferiority of SCTV01E-1 compared to SCTV01E. Furthermore, seroresponse rates (SRRs) of nAbs against Omicron BA.5 were 83.3% and 87.2% for the SCTV01E-1 and SCTV01E groups, respectively, with no marked difference observed (Appendix A).

##### Subgroup Analysis Based on Anti-SARS-CoV-2 Nucleocapsid Test

All COVID-19 vaccine-naïve participants did an anti-nucleocapsid protein (N-protein) test at baseline to ascertain their SARS-CoV-2 infection status pre-study vaccination. We categorized them into negative and positive N-protein antibody groups. The analysis indicated lower baseline GMTs of live virus nAb against Omicron BA.5 in participants in the negative N-protein group than in the N-positive group; however, their GMTs were numerically equivalent after the study vaccination.

Within the negative N-protein group, day 42 GMFRs over baseline reported by the negative group were 14.3 times (9.2, 22.2) and 35.6 times (22.7, 55.7) in the SCTV01E-1 and SCTV01E, respectively (Appendix A). The LS GMR between the SCTV01E-1 and SCTV01E groups was 1.01 (0.68, 1.51). SRRs of 90% (27/30) and 94.9% (37/39) were reported in SCTV01E-1 and SCTV01E, respectively. Day 178 GMFRs for SCTV01E-1 and SCTV01E groups increased by 1.4 (1.1, 1.7) times and 1.4 (1.1, 1.8) times, respectively, over day 150 (third dose) nAb levels. The LS GMR between the SCTV01E-1 and SCTV01E groups was 1.09 (0.65, 1.85). SRRs against Omicron BA.5 were 90.0% and 96.4% for SCTV01E-1 and SCTV01E, respectively.

Contrastingly, among the baseline positive N-protein group, day 42 GMFRs over baseline were 11.2 times (6.3, 19.8) and 8.0 times (4.6, 14.0) for SCTV01E-1 and SCTV01E groups, respectively (Appendix A). The LS GMR between the SCTV01E-1 and SCTV01E groups was 1.43 (0.84, 2.44). Day 42 SRR of live virus nAb against Omicron BA.5 was 88.0% (22/25) and 78.3% (18/23) in SCTV01E-1 and SCTV01E groups, respectively. Neutralizing antibody response on day 178 reported GMFRs for SCTV01E-1 and SCTV01E groups was 1.5 times (1.2, 1.8) and 1.4 times (0.9, 2.0), respectively, compared to day 150 antibody response (third dose). The LS GMR between the SCTV01E-1 and SCTV01E groups was 1.86 (1.07, 3.22). Day 178 SRRs against Omicron BA.5 were 75.0% and 63.6% for the SCTV01E-1 and SCTV01E groups, respectively.

#### 3.2.2. Cohort 2

##### Day 28 GMTs of Live Virus nAb and Seroresponse against Omicron BA.5

For the primary endpoint, 28 days after the first vaccination, the available immunogenicity data per i-PPS (immunogenicity per protocol set) were 224/240 (93.3%) participants: 112 (92.6%) in the SCTV01E-1 group and 112 (94.1%) in the SCTV01E group. Post-vaccination, GMTs of live virus neutralizing antibody response against Omicron BA.5 in both SCTV01E-1 and SCTV01E groups significantly increased over baseline (D0). The GMTs (95% CI) of nAb against Omicron BA.5 were 3543 and 3496 with corresponding GMFRs of 9.4× (95% CI: 7.3, 12.1) and 8.7× (95% CI: 6.5, 10.4) from baseline (D0) for SCTV01E-1 and SCTV01E, respectively (Figure 3). The LS GMR between the SCTV01E-1 and SCTV01E groups was 1.06 (0.86, 1.31), with LS GMR 95% LCI > 0.67, which met the pre-specified non-inferiority criteria. SRR of live virus nAb against Omicron BA.5 was 79.5% (89/112) and 79.5% (89/112) in the SCTV01E-1 and SCTV01E groups, respectively (Appendix A).

##### Day 148 GMTs of Live Virus nAb and Seroresponse Rates against Omicron BA.5

On day 148 (28 days after the second vaccination), available immunogenicity data per i-PPS were from 185 participants, with 96 in the SCTV01E-1 group and 89 in the SCTV01E group. On day 148, GMTs of anti-Omicron BA.5 variant nAb induced by SCTV01E-1 and SCTV01E were 3004 and 2603 with GMFRs (95% CI) of 1.7 times (1.5, 2.0) and 1.5 times (1.3, 1.8) over day 120 (before the second dose) for SCTV01E-1 and SCTV01E groups, respectively (Figure 3). The LS GMR of 1.17 (0.93, 1.47) between SCTV01E-1 and SCTV01E indicates nominal non-inferiority of SCTV01E-1 compared to SCTV01E. SRR of nAb against the Omicron BA.5 was also similar between SCTV01E-1 (76.9%) and SCTV01E (75.0%) (Appendix A).

Overall, after each study vaccination, a remarkable increase in neutralizing antibody titers was observed irrespective of baseline vaccination status and/or pre-existing immunity levels.

### 3.3. Safety

All randomized participants (n = 400) met the pre-specified eligibility criteria for safety analysis [Cohort 1; n = 160 and Cohort 2; n = 240].

In Cohort 1, the incidence of adverse events was comparable in both vaccine groups (Appendix A). The number and frequency of AEs after administration of SCTV01E-1 and SCTV01E were 13.6% (11/81) and 12.7% (10/79), respectively. Similar occurrences of solicited AEs were recorded in SCTV01E-1 and SCTV01E [(6.2%; 5/81) and [(7.6%; 6/79)], respectively. Likewise, SCTV01E-1 and SCTV01E reported comparable unsolicited AEs (11.1%; 9/81) and (10.1%; 8/79), respectively. Pain at the injection site and pyrexia were the most reported local and systemic AEs, respectively (Figure 4A,B). Most AEs were grade 1 or 2. Grade 3 or more AEs occurred < 2% in both vaccine groups, SCTV01E-1:1.2% (1/81), SCTV01E: 1.3% (1/79). Two IP-related ≥ grade 3 pyrexias were reported (one each in the SCTV01E-1 and SCTV01E groups). AEs were lower post-second and post-third doses than post-first dose. No deaths, SAEs, or AESI were reported during the follow-up period after each study vaccination.

Generally, for Cohort 2, SCTV01E-1 reported numerically higher incidences of AE than SCTV01E (Appendix A). The number and frequency of solicited AEs were 13.2% (16/121) and 10.9% (13/119) in SCTV01E-1 and SCTV01E, respectively. Unsolicited AE case distribution was 7.4% (9/121) and 6.7% (8/119) for SCTV01E-1 and SCTV01E, respectively. The most commonly reported local and systemic adverse events were pain at the injection site and pyrexia in both vaccine groups (Figure 4C,D). Numerically identical frequencies of ≥ grade 3 AEs were reported in SCTV01E-1 [1.7% (2/121)] and SCTV01E [1.7% (2/119)]. The SCTV01E-1 group reported two SAE cases, including one sensorineural hearing loss deemed vaccine-related AESI. The event resolved with treatment without any sequela. There were no deaths; no AEs led to study discontinuation.

## 4. Discussion

Despite cross-immunity among a vast majority of the global population [15], we are witnessing a worrying trend of constant Omicron sublineage re-infection in parallel with waning immunity [7,16,17]. Amidst all these, we investigated the safety and immunogenicity of Omicron-containing multivalent adjuvanted protein vaccines, SCTV01E-1 and SCTV01E, in two cohorts: unvaccinated and previously vaccinated populations, in a phase 2 trial. Our results provide comprehensive clinical immunogenicity data; as a primary series or booster, both SCTV01E-1 and SCTV01E induced markedly elevated GMTs of nAb levels and maintained for an extended period post-vaccination against Omicron BA.5 with matching tolerable safety profiles. The findings further validate our previous report of SCTV01E’s cross-reactive neutralizing antibodies against SARS-CoV-2 variants [12,13]. This is a testament to the fact that updated COVID-19 vaccines are needed in the quest to address waning immunity and strengthen the magnitude and durability of antibody responses against SARS-CoV-2 variants, considering first-generation vaccines reportedly have limited cross-neutralizing potency and breadth against evolving viral strains [18,19,20].

In the unvaccinated cohort, following the second dose, participants exhibited elevated GMTs (4930 and 4049) and GMFRs (12.8× and 20.5×) for SCTV01E-1 and SCTV01E, respectively, on day 42. While post-booster nAb levels (D178) were higher than pre-booster levels (D150), they were still lower than the nAb titers recorded on day 42 (Figure 2). The peak in nAb titers observed on day 42 could potentially be attributed to an antibody-ceiling effect as reported in other vaccine studies, albeit using different technologies [21]; nonetheless, nAb titers increasingly stabilized after booster vaccination (third dose) to offset any variability in nAb titers post-primary series. However, Cohort 1 nAb response dynamics should be interpreted cautiously as the target population consisted of two categories of unvaccinated participants: those with no infection history and those with prior SARS-CoV-2 infection.

To disentangle the influence of previous SARS-CoV-2 infection in our analysis of the anamnestic response to study vaccines, we used a nucleocapsid antibody test to establish a baseline (D0) SARS-CoV-2 antibody profile. Expectedly, higher baseline (D0) GMTs of nAb were observed in seropositive than seronegative participants. Both SCTV01E-1 and SCTV01E showed comparable nAb responses post-second dose, inducing high levels of GMTs and GMFRs in both seronegative and previously infected individuals. Third doses of both study vaccines also showed improved antibody response consistent with published literature [22,23] (Appendix A). Furthermore, a trial conducted in Qatar, a similar geographical region to the UAE, concluded that participants with previous SARS-CoV-2 infection and recent booster vaccination could induce a far superior immune response [24]. Additionally, our findings showed that although none of the participants in Cohort 1 reported previous SARS-CoV-2 infection, approximately 38.1% (61/160) tested positive for anti-nucleocapsid antibody, indicating asymptomatic SARS-CoV-2 infection. The proportion of asymptomatic COVID-19 cases observed in our trial is comparable to that of Al-Rifai et al., who reported 43.5% asymptomatic cases in the UAE [25]. With only 1.9% (3 out of 160) of participants having comorbid diseases, the relatively high percentages of asymptomatic cases could be due to the participants’ young age and absence of comorbidities [26]. Generally, SARS-CoV-2-specific IgG antibody levels in asymptomatic carriers are either negative or very low and will decrease over time. Therefore, asymptomatic carriers are still likely to benefit from vaccination. These findings demonstrate that unvaccinated individuals with or without SARS-CoV-2 infection could benefit from SCTV01E and SCTV01E-1 primary and/or booster dose vaccination.

We have previously reported the immunogenicity results of SCTV01E in healthy individuals primed with the inactivated COVID-19 vaccine [13]. Consistent with our earlier findings, in Cohort 2, peak GMTs (SCTV01E-1:3543 and SCTV01E:3496) against Omicron BA.5 were reached 28 days after the first dose, which is reflected in the high GMFRs of 9.4 (7.3, 12.1) times and 8.3 (6.5, 10.4) times from baseline (D0) for SCTV01E-1 and SCTV01E, respectively. Likewise, nAb GMTs increased 28 days after the second dose (Day 148) (Figure 3). Neutralizing antibody levels are highly predictive of immune protection [27] and typically peak approximately 4 weeks after vaccination [28]; we have shown that a longer dose interval is necessary for a two-dose regimen to maintain high immunogenicity response levels. However, our current study did not reveal any statistical difference between SCTV01E-1 and SCTV01E. A recent publication from other investigators provided evidence that (heterologous) SCTV01E booster vaccination could stimulate robustly higher immune responses against multiple SARS-CoV-2 variants, including Omicron sub-lineages like XBB and EG.5, than breakthrough infection, in addition to high levels of virus-specific memory B cells, thus conferring cross-neutralization protective immunity against future viral variants [29].

SCTV01E-1 and SCTV01E had clinically acceptable and comparable safety profiles as primary series and boosters in previously unvaccinated cohorts. No SAEs or AESIs were reported. Furthermore, the previously vaccinated cohort participants reported transient and mild AEs with short recovery times, consistent with our previously published clinical study results [12,13]. However, one rare vaccine-related AESI (deafness neurosensory) was reported in the SCTV01E-1 group. This event was resolved after receiving appropriate treatment. Neurosensory deafness is recognized for its idiopathic etiology, and it is noteworthy that current scientific evidence does not substantiate a causal relationship between COVID-19 vaccination and the development of sudden sensorineural loss [30].

This study has a few limitations. First, the trial participants’ characteristics are a replica of the population dynamics in UAE, which is relatively young and male-dominated; therefore, our findings may not be generalizable to the elderly population. We could not determine the elapsed time between previous SARS-CoV-2 infection and study vaccination in the previously unvaccinated positive N-protein subgroup due to practical challenges. Finally, nAb against other SARS-CoV-2 variants, especially XBB and its sub-lineages, were not tested.

## 5. Conclusions

Our results provide critical immunological insights to inform public health strategies in the context of vaccine policy decisions, including the need for boosting with multivalent vaccines. Both SCTV01E-1 (with BA.5 antigen strain) and SCTV01E (without BA.5 antigen strain) were highly immunogenic against Omicron BA.5 in both unvaccinated and previously vaccinated individuals, and neutralizing antibody titers persisted for long duration at highly effective levels.

## Figures and Tables

**Figure 1 vaccines-12-01109-f001:**
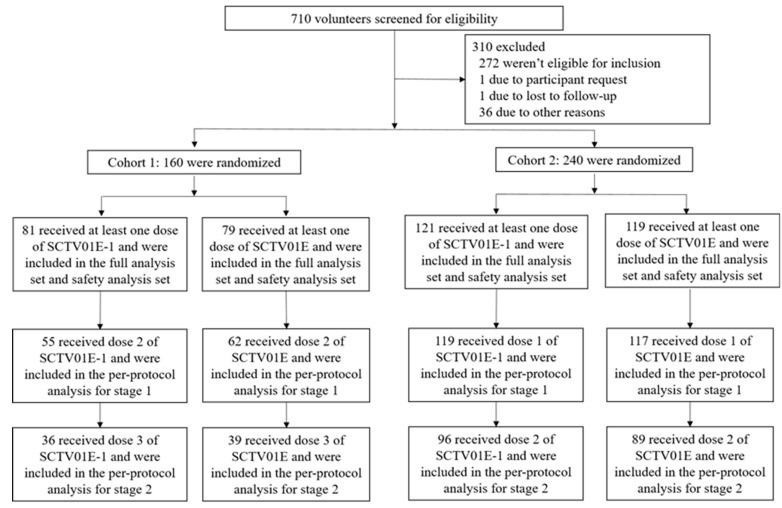
Randomization and Analysis Populations.

**Figure 2 vaccines-12-01109-f002:**
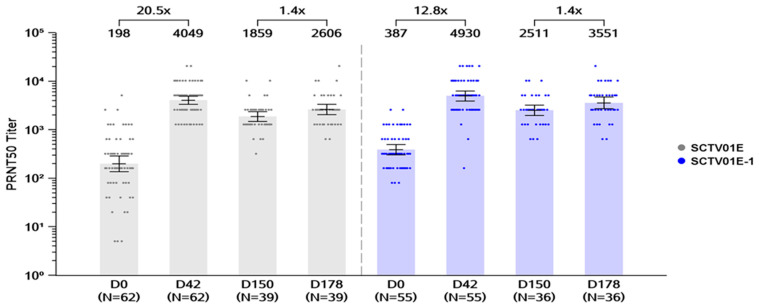
GMTs of neutralizing antibodies against BA.5 in the unvaccinated cohort. GMTs of neutralizing antibodies were measured using a 50% plaque reduction neutralization test (PRNT50). Bars show the GMTs on days 0, 42, 150, and 178. The center of the bars represents the GMT. Dots represent the values of individual participants. Note: SCTV01E group (grey), SCTV01E-1 group (blue). Abbreviations: GMT, geometric mean titer; PRNT50, 50% plaque reduction neutralization test.

**Figure 3 vaccines-12-01109-f003:**
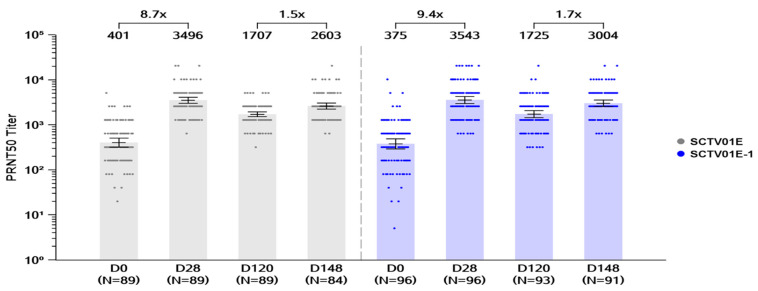
GMT of neutralizing antibodies against BA.5 in the previously vaccinated cohort. GMTs of neutralizing antibodies were measured using a 50% plaque reduction neutralization test (PRNT50). Bars show the GMTs on days 0, 28, 120, and 148. The center of the bars represents the GMT. Dots represent the values of individual participants. Note: SCTV01E group (grey), SCTV01E-1 group (blue). Abbreviations: GMT, geometric mean titer; PRNT50, 50% plaque reduction neutralization test.

**Figure 4 vaccines-12-01109-f004:**
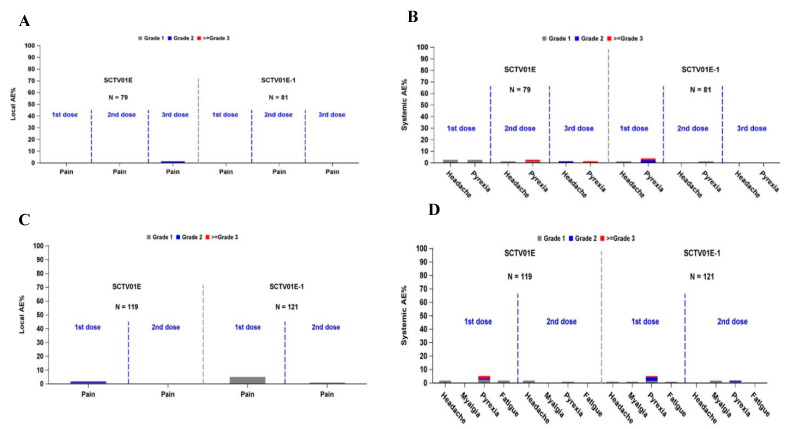
Safety Summary: (**A**). Local AEs in Cohort 1; (**B**). Systemic AEs in Cohort 1; (**C**). Local AEs in Cohort 2; (**D**). Systemic AEs in Cohort 2. AE, Adverse event.

**Table 1 vaccines-12-01109-t001:** Baseline clinical characteristics in Cohort 1.

	SCTV01E (N = 79) n (%)	SCTV01E-1 (N = 81) n (%)	Total (N = 160) n (%)
**Age (years)**			
n	79	81	160
Mean (SD)	27.9 (7.36)	29.1 (8.32)	28.5 (7.86)
Median	26.0	26.0	26.0
Min, Max	18, 48	18, 57	18, 57
**Age Group, n (%)**			
18–54	79 (100.0)	80 (98.8)	159 (99.4)
≥55	0	1 (1.2)	1 (0.6)
**Gender, n (%)**			
Female	1 (1.3)	0	1 (0.6)
Male	78 (98.7)	81 (100.0)	159 (99.4)
**Race, n (%)**			
American Indian or Alaska Native	0	1 (1.2)	1 (0.6)
Asian	74 (93.7)	74 (91.4)	148 (92.5)
Black or African American	2 (2.5)	2 (2.5)	4 (2.5)
White	2 (2.5)	1 (1.2)	3 (1.9)
Not Reported	1 (1.3)	2 (2.5)	3 (1.9)
Unknown	0	1 (1.2)	1 (0.6)
**BMI (kg/m^2^)**			
n	79	81	160
Mean (SD)	22.94 (3.342)	23.65 (3.639)	23.30 (3.503)
Median	22.50	23.30	22.90
Min, Max	16.8, 30.5	17.1, 32.8	16.8, 32.8
**Anti-SARS-CoV-2 nucleocapsid antibody test, n (%)**			
Positive	30 (38.0)	31 (38.3)	61 (38.1)
Negative	49 (62.0)	50 (61.7)	99 (61.9)

N, number; SD, Standard Deviation; BMI, Body Mass Index.

**Table 2 vaccines-12-01109-t002:** Baseline clinical characteristics in Cohort 2.

	SCTV01E (N = 119) n (%)	SCTV01E-1 (N = 121) n (%)	Total (N = 240) n (%)
**Age (years)**			
n	119	121	240
Mean (SD)	28.2 (6.61)	28.3 (7.30)	28.2 (6.95)
Median	26.0	26.0	26.0
Min, Max	19, 54	18, 59	18, 59
**Age Group, n (%)**			
18–54	119 (100.0)	120 (99.2)	239 (99.6)
≥55	0	1 (0.8)	1 (0.4)
**Gender, n (%)**			
Female	1 (0.8)	1 (0.8)	2 (0.8)
Male	118 (99.2)	120 (99.2)	238 (99.2)
**Race, n (%)**			
Asian	119 (100.0)	120 (99.2)	239 (99.6)
Black or African American	0	1 (0.8)	1 (0.4)
**BMI (kg/m^2^)**			
n	119	121	240
Mean (SD)	23.75 (4.173)	23.72 (3.877)	23.73 (4.018)
Median	23.50	23.50	23.50
Min, Max	16.4, 36.0	16.2, 37.2	16.2, 37.2
**Number of previous COVID-19 vaccines, n (%)**			
2	107 (89.9)	109 (90.1)	216 (90.0)
3	12 (10.1)	12 (9.9)	24 (10.0)
**Time since last COVID-19 vaccination (month), n (%)**			
3–5	1 (0.8)	0	1 (0.4)
6–24	118 (99.2)	121 (100.0)	239 (99.6)
**COVID-19 Vaccine name, n (%)**			
Sinopharm Inactivated COVID-19 Vaccine	88 (73.9)	81 (66.9)	169 (70.4)
CoronaVac	31 (26.1)	40 (33.1)	71 (29.6)
**Diagnosed with COVID-19, n (%)**			
Yes	3 (2.5)	5 (4.1)	8 (3.3)
No	116 (97.5)	116 (95.9)	232 (96.7)

N, number; SD, Standard Deviation; BMI, Body Mass Index.

## Data Availability

The sponsor, investigator, and other collaborators will review and approve data-sharing proposals directed to the corresponding author based on scientific merit. Anonymized participant data will be made available and shared through a secure online platform after signing a data access agreement.

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
