# Peer review of "Immunogenicity and Safety of Omicron-Containing Multivalent COVID-19 Vaccines in Unvaccinated and Previously Vaccinated Adults"

_vaccines, 2024, doi:10.3390/vaccines12101109_

Round 1
Reviewer 1 Report
Comments and Suggestions for Authors The authors address an important topic in the ongoing evolution of COVID-19 vaccines, particularly focusing on the immunogenicity and safety of two multivalent vaccines, SCTV01E and SCTV01E-1, against the Omicron BA.5 variant. Overall, the manuscript is well written. However, there are some issues.- Review the manuscript carefully, focusing on the correspondence between the absolute frequency and the percentage frequency. For example, "29.6% (71/230)" (line 201) or "11.1% (9/121) and 10.1% (8/119)" (line 330). Please check, they seem incorrect
- Verify spacing errors. For example, "success[4]" (line 44).
- The caption of Figure 4 missed the Cohort number about the Figure 4B. Please replace "Systemic AEs in Cohort" with "Systemic AEs in Cohort 1"(line 324).
- The subsection 3.1 is repeated twice: "3.1. Study Participants, Demographics and Baseline Characteristics" and "3.1. Immunogenicity". Therefore, it would be appropriate to review the formatting and edit the numbering of sections and subsections accordingly.
- Verify and standardize the references reported as link. For example, bibliographical reference n.8 reports "; Available from:" while it is absent for references n. 9 and n.10.
- It would be useful to provide a list of acronyms used in the manuscript.
Reviewer 2 Report
Comments and Suggestions for Authors
This is an important study, well planned and conducted.
Introduction
“The study vaccines, SCTV01E [Alpha/Beta/Delta/Omicron 63 (BA.1)] and SCTV01E-1 [Alpha/Beta/Delta/Omicron (BA.1/BA.4/5)] have similar antigen 64 composition except BA.4/5 antigen strain”
[this is a very brief description. Please fully describe the vaccines so that readers can understand. If this is an in house trial by the Chinese manufacturer, if you are going to publish in a recognised journal you must make the detailed composition of your vaccine known].
Materials and methods
[The study was performed in Dubai. Please explain why this location was chosen and not one in China. Have these vaccines been tested in other locations or in in house trials?]
Statistical Analysis
“The design was to achieve at least 80% power to demonstrate that SCTV01E-1 was 159 superior to SCTV01E in terms of the GMTs of neutralizing antibody against Omicron BA.5 160 with a one-side type 1 error of 0.025. Cohort 1 sample size was determined based on the 161 following assumptions: nAb GMT ratio of SCTV01E-1 versus SCTV01E ≥ 1.7; the standard 162 deviation of nAb under log10 transformation is 0.45; the dropout rate is about 10%. Like- 163 wise, Cohort 2 on the following assumptions: nAb GMT ratio of SCTV01E-1 vs SCTV01E 164 ≥ 1.5; the standard deviation of nAb under log10 transformation is 0.45; the dropout rate 165 is about 10%.”
[The precise origins of the power computation are important. Where do the data for standard deviation for nAb come from?]
Study population
[Why limited to 8-54]
[Why no females?]
Results
“the avail- 217 able immunogenicity data per i-PPS (immunogenicity per protocol set) was for 117/160 218 participants: 55 in the SCTV01E-1 group and 62 in the SCTV01E group.
[Why were immunogenicity data missing for the stated individuals. How does this affect the generalisability of your results?]
Conclusions
“Both SCTV01E-1 (with BA.5 antigen strain) and SCTV01E (without BA.5 antigen 411 strain) were highly immunogenic against Omicron BA.5 in both unvaccinated and previ- 412 ously vaccinated individuals and neutralizing antibodies titers persisted for long duration 413 at highly effective levels.”
[Can you please explain why the vaccine without BA.5 antigen was equally effective?]
Reviewer 3 Report
Comments and Suggestions for Authors
Authors, this is a good and thorough paper examining the impact of vaccination on individuals with prior exposure to COVID and those with no exposure or at least no detectable exposure to this virus.
You have presented data in a reasonable format, although there are many acronyms, that makes interpretation confusing at times. Perhaps a supplemental table that lists each acronym could be helpful.
It is unfortunate that you did not have more late aged cohorts in your study, since these individuals seem most susceptible to COVID and the long term effects of the virus.
Overall a good paper.
Reviewer 4 Report
Comments and Suggestions for Authors
This phase 2 RCT assessed the immunogenicity and safety of multivalent COVID-19 vaccines, SCTV01E and SCTV01E-1 against Omicron BA.5. Overall, this study showed the promising effect. However, I have two minor concerns.
1. The immune status of included subjects should be mentioned.
2. The effect of previous COVID-19, particularly for asymptomatic infection should be discussed.
